# Ultrafiltration Membranes Modified with Reduced Graphene Oxide: Effect on Methyl Green Removal from Aqueous Solution

**DOI:** 10.3390/ma16041369

**Published:** 2023-02-06

**Authors:** María Dolores Murcia, Asunción M. Hidalgo, María Gómez, Gerardo León, Elisa Gómez, Marta Martínez

**Affiliations:** 1Departamento de Ingeniería Química, Campus de Espinardo, Universidad de Murcia, 30100 Murcia, Spain; 2Departamento de Ingeniería Química y Ambiental, Universidad Politécnica de Cartagena, Paseo Alfonso XIII 52, 30206 Cartagena, Spain

**Keywords:** methyl green, modified membranes, graphene oxide, reduced graphene oxide, ultrafiltration

## Abstract

In this work, three types of ultrafiltration membranes with different characteristics (GR60PP, RC70PP and GR80PP) have been tested for the removal of the dye methyl green. The tests were first carried out with the three membranes without any modification and then with the membranes’ surfaces modified with reduced graphene oxide (rGO). The modification was achieved through physical treatment. The CR70PP membrane did not support the modification treatment and was discarded. The other membranes were initially characterized with distilled water tests to study the permeability to the solvent, and later, the permeate fluxes and the values of rejection coefficients were obtained at different working pressures with a fixed dye initial concentration. In addition, SEM images and SEM-EDX spectra of the native and modified membranes were obtained before and after the dye tests. The GR60PP membrane has shown the best results in relation to the modification because it has increased its rejection levels. On the opposite, the GR80PP membrane performs better without surface modification, achieving the highest rejection values and the highest permeate fluxes in its native form.

## 1. Introduction

Membrane technology use has gained special importance in recent decades since the increase in the world population has required an improvement in water reuse treatments. The world population has grown by around 21% in the last 17 years, going from 6.450 million people to approximately 7.800 million. Regarding water consumption, the amount needed for world consumption has increased six times since the beginning of the 20th century, as overexploitation, pollution, and the effects of climate change have increased. Almost 40% of terrestrial human beings have problems of water shortage, and a current consumption of 54% of fresh water available on the planet is estimated; about 3.600 km^3^ of fresh water is extracted for this consumption, of which approximately half is not consumed, and of the other half, there is an estimated 65% dedicated to agriculture, 25% to industry, and 10% to households [1].

It should be noted that membrane technology is widely used due to the advantages it presents at economic, energy and handling levels. These processes can be classified based on the operating pressure and molecular cut-off size of the membranes in microfiltration, ultrafiltration, nanofiltration and reverse osmosis. In recent years, ultrafiltration membrane processes have gained popularity because they can remove most organic molecules and viruses. In addition to producing a stable quantity of water regardless of its origin, they remove most of the pathogens, and the use of chemical products is not necessary, except for cleaning the membranes [2]. Although these membranes have many advantages that make them a useful research target, they have some drawbacks, such as fouling, which requires necessary maintenance for the correct operation of the filtration process. For this reason, the modification of the membranes has become an important object of study in terms of improving their filtration properties.

One of the advantages of using membrane technology as a separation technology is that it works without the addition of chemicals, involving relatively low energy use. The use of membranes is increasingly widespread for the treatment of groundwater, surface or wastewater. The applications of membrane technology range from the pharmaceutical, nautical, agri-food, electronic and nuclear industries to the treatment of water and industrial effluents, the desalination of seawater and medicine [3]. In this work, polymeric membranes have been used since another of the advantages that they present is the lower economic cost compared to other separation methods. Furthermore, this method allows the use of products of different viscosities that would be difficult to filter by other filtration methods. It is also an advantage to obtain a cleaner final product of higher quality with a lower cost of cleaning [3]. In addition to the advantages that have just been mentioned, ultrafiltration membranes can also be more useful than other types of membranes, depending on what is sought in the experiment, since they require lower operating pressures, do not present phase changes, and the optimum operating temperature is relatively low [4]. 

Polysulfone is a widely used compound for ultrafiltration membranes, as it has high thermal and chemical stability. On the other hand, the hydrophobic surface of these membranes gives rise to fouling when exposed to protein solutions. This fouling reduces the flow of the membrane, requiring greater economic expense in the cleaning process. MK Sinha and M.K. Purkait [5] studied the modification of polysulfone ultrafiltration membranes with a poly (N-vinylcaprolactam-co-acrylic acid) copolymer (poly (VLC-co-AA)) as an additive, observing an increase in permeate flux and a decrease in fouling in the modified membranes. In turn, Hong Zhao et al. [6] comment on the decrease in fouling in a type of polypropylene membrane thanks to the modification of its hydrophobic surface to achieve a more hydrophilic surface. These authors carried out the modification with a zwitterionic polymer (poly(sulfobetaine methacrylate) [poly(SBMA)]). Research carried out with inorganic compounds for modifications must also be considered, such as the study by Feng Li et al. [7], which uses titanium dioxide (TiO_2_) nanoparticles on polysulfone-based membranes. The results of that study demonstrate an improvement in permeate flux, fouling, and mechanical strength of the membranes after modification.

However, there are still limitations regarding the number of nanoparticles that adhere to the surface of the membranes through the different modification methods studied. The tendency for ultrafiltration membranes to foul is strongly connected with the charged groups on the surface of the polymers and with the charged groups resulting from contact with the compounds that pass through them. But the studies qualitatively show that if the surface of the membrane is hydrophilic and the molecules of the solution that cross it are of the same charge, the tendency to fouling decreases [8].

Carbon nanomaterials, both nanotubes and graphene, have been used in recent years as nanofillers to improve the performance of separation membranes. However, these compounds are hydrophobic, which means that they cannot improve the resistance of membranes against fouling. On the other hand, graphene oxide (GO) contains polar groups (hydroxyl and epoxy functional groups) in its basal planes and carbonyl and carboxyl groups at the edges. This implies that this oxide can be dispersed more easily in a polymeric matrix since its functional polar groups interact more strongly with the polymeric chains. Therefore, when GO is used as a filler for polymeric membranes, it is expected to improve mechanical strength, thermal stability and hydrophilicity despite the hydrophobic nature of these membranes. In recent years, there have been several studies that demonstrate the benefits of using GO and rGO compounds. For example, Ghosh et al. [9] studied a new reduced graphene oxide (rGO) synthesis route that results in a highly conductive membrane without using organic solvents or binders.

Methyl green dye was chosen for this research due to its molecular size as a pattern to know the improvement capacity of the membranes. The study of the phenomenon of membrane modification with reduced graphene oxide is important. If a membrane that easily retained the molecules of this dye had been chosen, it would not be possible to appreciate whether or not the membranes improved with the modification. To date, numerous investigations have been carried out on membrane modification techniques, and it has also been shown that graphene oxide has remarkable mechanical and resistance properties, which makes it susceptible to being studied in depth. Therefore, the objective of this study was the use of reduced graphene oxide to modify different ultrafiltration membranes and observe their behavior when a dye (methyl green) passes through them.

## 2. Materials and Methods

### 2.1. Materials

#### 2.1.1. Reagents

Methyl green zinc chlorine salt, C_27_H_35_BrClN_3_·ZnCl_2_, molecular weight 653.24 g/mol, supplied by Sigma-Aldrich (Barcelona, Spain), and reduced graphene oxide (80% C) was obtained from Abalonyx (Oslo, Norway).

#### 2.1.2. Membranes

The three ultrafiltration membranes used in this study were supplied by Alfa Laval (Madrid, Spain). Their main technical characteristics are shown in Table 1. 

### 2.2. Equipment

#### 2.2.1. Membrane Test Module

The equipment used for the tests is a Triple System Model F1 membrane module from the commercial company MMS, Urdorf (Zurich, Switzerland). It is designed for a maximum operating pressure of 40 bar. The feed tank is made of steel, is closed to the atmosphere, and has a maximum capacity of 8 × 10^−3^ m^3^. Three membranes with a circular area (2.8 × 10^−3^ m^2^) can be inserted into the membrane module [10].

#### 2.2.2. Spectrophotometer 

The model of the spectrophotometer is an Evolution 300 visible-ultraviolet from Thermo Electron, Thermo Fisher Scientific (Waltham, MA, USA). It has a xenon lamp platform and a variable bandwidth [11]. To measure the samples introduced into the spectrophotometer, a 1 cm optical path quartz cuvette with a 3 mL capacity was used.

#### 2.2.3. Variable Pressure Scanning Electron Microscopy (SEM) 

A variable pressure SEM scanning electron microscope, model HITACHI S-3500N (Hitachi High-Technologies Corporation, Tokyo, Japan), was used. This machine has a resolution of 3 nm (high vacuum mode) or 4.5 nm (low vacuum mode). Its pressure range can vary from 1 to 270 Pa, and it has a digital image resolution of up to 2560 × 1920 pixels [12]. It detects secondary electrons, variable pressure secondary electrons, and Robinson backscattered electrons. It is equipped with an EDX XFlash 5010 analysis system X-ray detector (Brukers AXS, Karlsruhe, Germany) [13].

### 2.3. Methods

#### 2.3.1. Membranes Modification

The reduced graphene oxide solution is introduced into a Branson 450D sonicator (Emmerson Ultrasonic Corporation, Dansbury, CT, USA), equipped with a flat tip probe of 1.27 cm in diameter at an amplitude of 30% in pulses of 15 s ON and 15 s OFF for 10 min. Sonicator tests are performed to improve the adherence of rGO to the surface of the membranes. The membrane is then placed in a funnel with the active side up, and the rGO solution is passed through vacuum filtration.

#### 2.3.2. Morphological Characterization of Membrane through SEM and EDX

Membrane samples have been scanned with scanning electron microscopy (SEM) and X-ray energy dispersion spectroscopy (EDX), obtaining images of the membrane surfaces from 70 µm to 800 µm and at an acceleration voltage of 15 kV. 

#### 2.3.3. Methyl Green Analysis

For the analysis of the dye (methyl green) in the feed and permeate samples, a calibration curve was made. Initially, an absorption spectrum of methyl green was carried out in the Evolution 300 spectrophotometer, and it was found that the maximum absorption wavelength was 635 nm. All the experiments were done in duplicate, and the average standard deviation was below 4%.

### 2.4. Physico-Chemical Characterization of the Membrane. Characteristic Parameters of the System

In order to determine the separation capacity of the membranes and to know if there was an improvement in filtration with the modification of reduced graphene oxide, three characteristic parameters have been studied, water permeability, permeate flux and rejection coefficient. 

For the determination of the permeability of the membranes, they were immersed in a mixture of tap water and distilled water to activate their active surface. After this, distilled water is passed through them at pressures between 3 and 10 bar, depending on the type of membrane. The tests cover a duration of up to 40 min, also depending on the type of membrane, since the permeate flux is higher for 1 of the types of membranes used, and this causes the feed tank to empty more quickly. The water permeability coefficient (*A_w_*) was obtained by the equation:*J_w_* = *A_w_* (∆*P* − ∆*Π*)(1)
where *J_w_* is the solvent permeate flux (kg/m^2^ s), *A_w_* is the solvent permeability coefficient (s/m), and Δ*P* and Δ*Π* are operating and osmotic pressure, respectively (Pa). *A_w_* can be determined as the slope of the representation of *J_w_* versus Δ*P*. 

Permeate fluxes (*J_p_*) were determined by the following equation:(2)$$J_{P}=\frac{Q_{P}}{S}$$
where *J_p_* (kg/(m^2^ s) is the permeate flux, *Q_p_* is the mass flow rate (kg/s), and *S* is the active membrane area (m^2^)

On the other hand, the rejection coefficient is defined as the ratio between the concentration of the feed solution minus that of the permeate and the concentration of the feed solution. This parameter expresses the capacity of the membrane for dye removal.
(3)$$R\%=\frac{Ca-Cp}{Ca\;}\;100$$

To calculate this parameter, the experimental tests have been carried out with a 1 g/L solution of methyl green in water. The dye was passed through the native and modified membranes after the first tests with distilled water. These assays have been carried out once for each type of membrane used, with and without the modification of the reduced graphene oxide, except for the RC70PP, since this membrane has not supported the modification.

#### Anti-Fouling Test

The water flux permeability ratio *F_w_* was used to characterize the percentage of the used ultrafiltration membranes (native and modified) reaching the initial level after cleaning.

Usually, different authors have calculated this parameter using the initial and post-treatment water flux [10,14,15,16]. In this work, the coefficients of permeability to the solvent (water) have been used initially and after the dye has passed through the membrane.
(4)$$F_{w}=\frac{A_{wf}}{A_{wi}}\;\cdot 100$$

## 3. Results and Discussion

### 3.1. Morphological Characterization of the Membranes

This section shows the images and spectra obtained in the SEM scanning electron microscopy and in the SEM-EDX spectra to know if the modifications made in the membranes have been effective at microscopic levels. Figure 1 shows the SEM images of the native and modified GR60PP and GR80PP membranes. As was previously commented, the RC70PP membrane was not modified with rGO, so it was discarded for further tests. This membrane is made of regenerated cellulose acetate in polypropylene, and it has little tolerance to aqueous solutions. In the bibliography, it is shown that cellulose acetates are susceptible to being hydrolyzed [17].

Figure 1 shows that the coating of both membranes results in an increase in their superficial roughness due to the interfacial enrichment of reduced graphene oxide onto the membrane, and this effect is more pronounced for the GR60PP. 

Figure 2 depicts the SEM-EDX spectra of the native and modified GR60PP and GR80PP membranes. For the GR60PP membrane, more elements can be seen in the spectra after the modification with reduced graphene oxide, which matches the SEM results and confirms that the surface modification is more noticeable for this membrane. 

The comparison of the percentages of carbon and oxygen obtained from the quantitative analysis of the GR60PP and GR80PP membranes’ EDX spectra, both native and rGO-modified, shows an increase in the C/O ratio in the modified membranes with respect to the natives, which demonstrates the retention of rGO on the surface of those modified membranes. This C/O ratio increment is 2.66 times in the GR60PP membrane and 1.79 times in the GR80PP membrane, proving once again that the modification is more effective in the case of the GR60PP membrane.

### 3.2. Physico-Chemical Characterization of the Membranes

#### 3.2.1. Solvent Permeability

Figure 3 represents the water permeate flux *J_w_* against the hydraulic pressure gradient (∆P) before the dye experiments with the values of each native and modified membrane.

Figure 3 shows that the highest permeate flux of water is obtained with the native GR80PP membrane, while the rest of the membranes, both native and modified, present smaller values of permeate flux. For all the membranes, it is common that the amount of collected permeate increases with increasing pressure. According to the research of Ravishankar et al. [18], the permeate flux of the solvent (water) increases with the concentration of graphene oxide in the membrane. In this research, the permeate flux of water was lower with the presence of reduced graphene oxide in the case of the GR80PP membrane, while, for the GR60PP, there was a slight increase in permeate flux. So the surface modification seems to have worked better for the GR60PP membrane. Table 2 shows the permeability coefficients of each membrane obtained from the fittings of Figure 3, confirming that the solvent permeability improves after membrane modification only in the case of the GR60PP membrane. 

In order to compare the values of the permeability coefficients obtained for each membrane with the data collected in the literature, the following studies are discussed. In work carried out by Sánchez–Moya et al. [19], in which they experimented with the GR60PP and GR80PP membranes in the same range of operating pressures that are studied in this research to separate lactose and protein from sheep milk soil, they obtained *A_w_* values of 6.69 × 10^8^ s/m for the GR60PP membrane and 5.38 × 10^8^ s/m for the GR80PP. Comparing these values with those obtained experimentally in this work, a great similarity is observed with respect to the permeability coefficient of the GR60PP membrane. While for the GR80PP, there is a more than two-fold difference, being higher than the data obtained in this experiment.

#### 3.2.2. Methyl Green Removal

Figure 4 depicts the relationship between the permeate fluxes of the dye solution for native and modified membranes and the tested operating pressures. 

It can be seen how the native GR80PP membrane is the one with the highest permeate flux, followed by the modified GR80PP. The membrane with the lowest permeate fluxes is the GR60PP, both native and modified.

Comparing Figure 3 and Figure 4, it can be seen that the permeate fluxes of water are higher than the permeate fluxes of dye for the native membranes. However, the modified membrane presents higher values of dye flux than water flux, particularly the modified GRP80PP membrane. 

In similar studies, Ravishankar et al. [18] tested the pass of a lead nitrate solution through different membranes, verifying that the flow increases linearly with pressure. However, graphene oxide-modified membranes exhibit lower fluxes for lead dissolution than for pure water. This could be due to an accumulation of osmotic pressure caused by the nitrate salt retained in the graphene oxide of the membrane. Furthermore, Macedo et al. [20] studied the role of concentration polarization in the ultrafiltration of sheep milk whey with the use of three different ultrafiltration membranes. This study concluded that the fluxes of pure water through the membranes are always much greater than the fluxes of permeate of sheep serum, which suggests that at the beginning of the experiment, there is a fouling of the membrane that causes the rapid decrease in permeate flux, which is confirmed by the high value of resistance to fouling that they calculate. Of the three membranes used, the one with the highest permeate flux is the one that turns out to be the most resistant to fouling because the concentration of polarization controls mass transfer.

Figure 5 shows the values of the rejection coefficient of native and modified membranes versus the operating pressures.

As can be seen in Figure 5, the rGO GR80PP modified membrane is the one with the worst rejection coefficients, and there is a significant difference with the native membrane, which shows high rejection values. It proves once again that, for this membrane, the modification with reduced graphene oxide seems to have worsened its filtration results. On the other hand, the rejection coefficient values increased considerably when the GR60PP membrane was modified; so for this membrane, the modification has improved the dye selectivity of separation efficiency.

#### 3.2.3. Fouling Study and Membrane Deterioration

To study the behavior of the membranes after their use and check if there is any fouling or membrane deterioration, the solvent permeability tests were repeated. Figure 6 represents the water permeate flux *J_w_* against the hydraulic pressure gradient (∆P) after the dye-passing with the values of each of the native and modified membranes. 

Table 3 shows the permeability coefficients of the membranes after the dye-passing for native and modified membranes calculated from the fittings of Figure 6.

Comparing the values presented in Table 2 with those presented in Table 3, it can be seen that the solvent permeability coefficient decreases for native membranes and increases for modified membranes. 

In the research work carried out by Sánchez–Moya et al. [19], where the same membranes were used, the whey permeate flux was lower than the water flux, suggesting that the adsorption fouling and blockage of the membrane pores was quite significant.

According to the literature, the whey permeate flux is lower than the water flux. When solute molecules are smaller than or similar to the pore size of membranes, these molecules can penetrate into membrane pores, gradually reducing their effective radius or causing the pore to become completely blocked. Membrane fouling is more noticeable when the difference between the pore size and the molecular size of the solute in the solution used is smaller [19,20,21,22]. Comparing the experimental results, it can be seen that the initial water flux is higher than the final water flux in the non-modified membranes and, on the other hand, the initial water flux is lower than the final water flux in the modified membranes. So, in good agreement with the literature, native membranes are more susceptible to fouling than modified ones.

As described above, the GR60PP membrane that presents a larger MWCO (25,000 Da) than GR80PP (10,000 Da) will be more prone to fouling. This can be seen in the comparison of the *A_w_* values in Table 2 and Table 3. The decrease in *A_w_* after the dye-passing is greater for the native GR60PP membrane than for the native GR80PP.

However, the results show that the permeability coefficient increased with the modification. There is no decrease in the permeate flux after the passage of the dye, so the effect of fouling with the presence of reduced graphene oxide could be neglected. This phenomenon could be due to an opening of the membrane pores. But, considering Figure 5, a considerable increase in the dye rejection coefficient is observed when comparing the native GR60PP with the modified one, the latter being the one that performs the best solute separation. These results confirm what many studies have previously described, the performance improvement of rGO when used for nanofiltration and ultrafiltration membranes [23,24,25].

## 4. Conclusions

SEM images and SEM-EDX spectra showed that the membrane surface modification with reduced graphene oxide was more efficient for the GR60PP than for the GR80PP membrane. Also, there is an increase in the C/O ratio in the modified membranes with respect to the native ones, which confirms the surface modification, being this increase more significant for the GR60PP membrane. For the initial characterization tests of all the membranes with distilled water, the highest flux values were obtained with the native GR80PP membrane, while the lowest were similarly achieved with the modified GR60PP and GR80PP membranes. The GR60PP membrane slightly improved its solvent permeability after rGO modification. With regard to dye tests to define the selectivity of the membranes, the highest permeate flux again corresponded to the native GR80PP membrane. But, this time, the fluxes of the modified membranes were greater than those for distilled water. As for the dye rejection coefficient obtained, the highest values were reached with the native GR80PP. Comparing the rejection between native and modified membranes, the GR60PP rejection increased considerably with the reduced graphene oxide modification. With the initial and final characterization of the membranes, a decrease in the water permeate flux for the native membranes was observed, as expected in membrane fouling. However, for the modified membranes, the opposite effect was appreciated: the permeate solvent flux increased. Further research is needed to explain the observed behavior.

As a final conclusion, in order to improve the process with the reduced graphene oxide modification, the GR60PP membrane has been shown to be the best option. Meanwhile, when working with the native membrane, the GR80PP is the one that obtains the best results.

## Figures and Tables

**Figure 1 materials-16-01369-f001:**
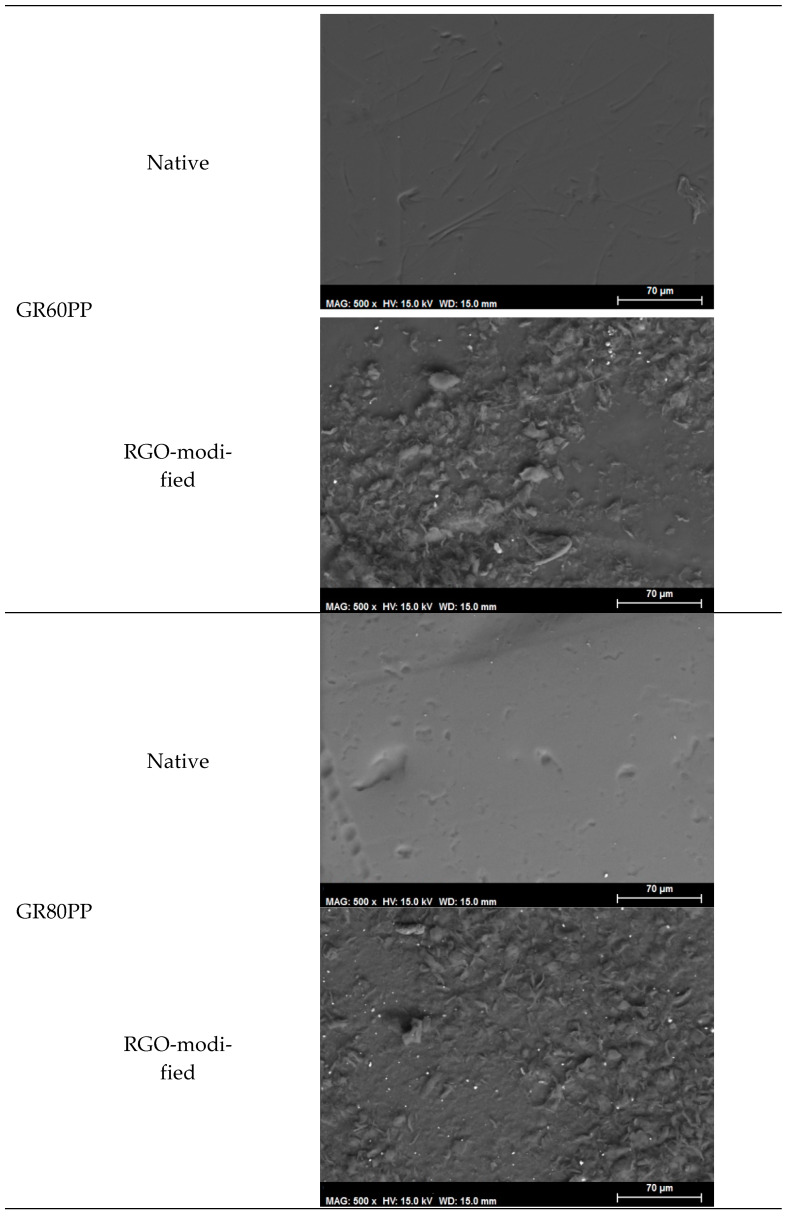
SEM images of the membranes (before the dye step).

**Figure 2 materials-16-01369-f002:**
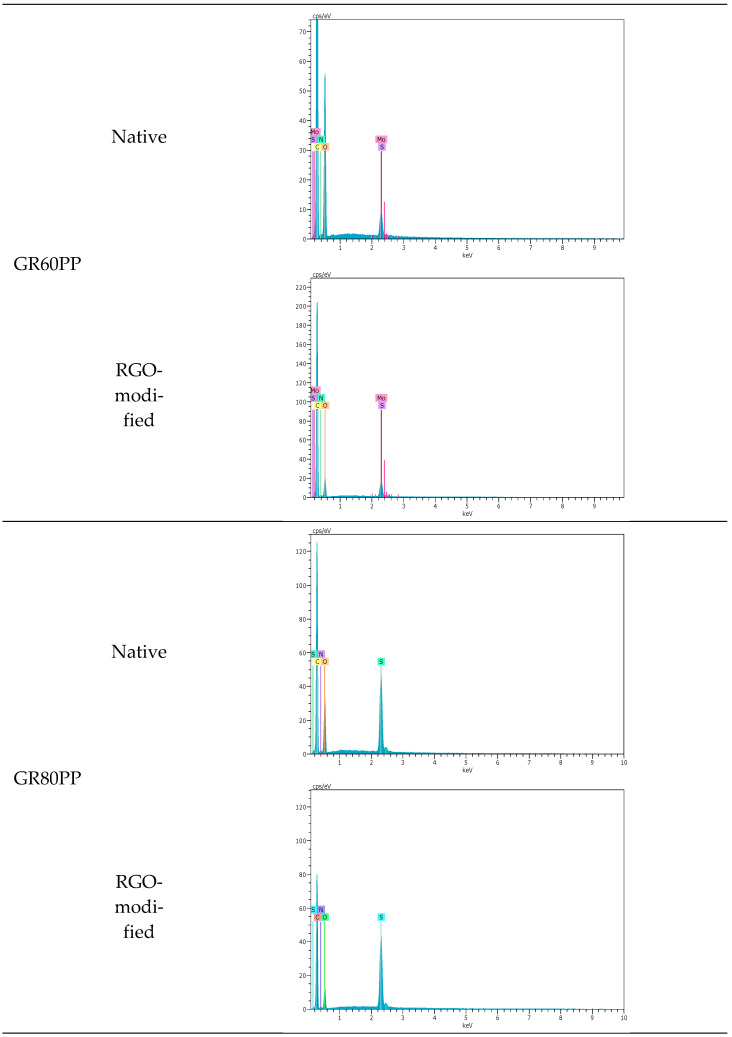
SEM-EDX spectra for the different membranes (before the dye step).

**Figure 3 materials-16-01369-f003:**
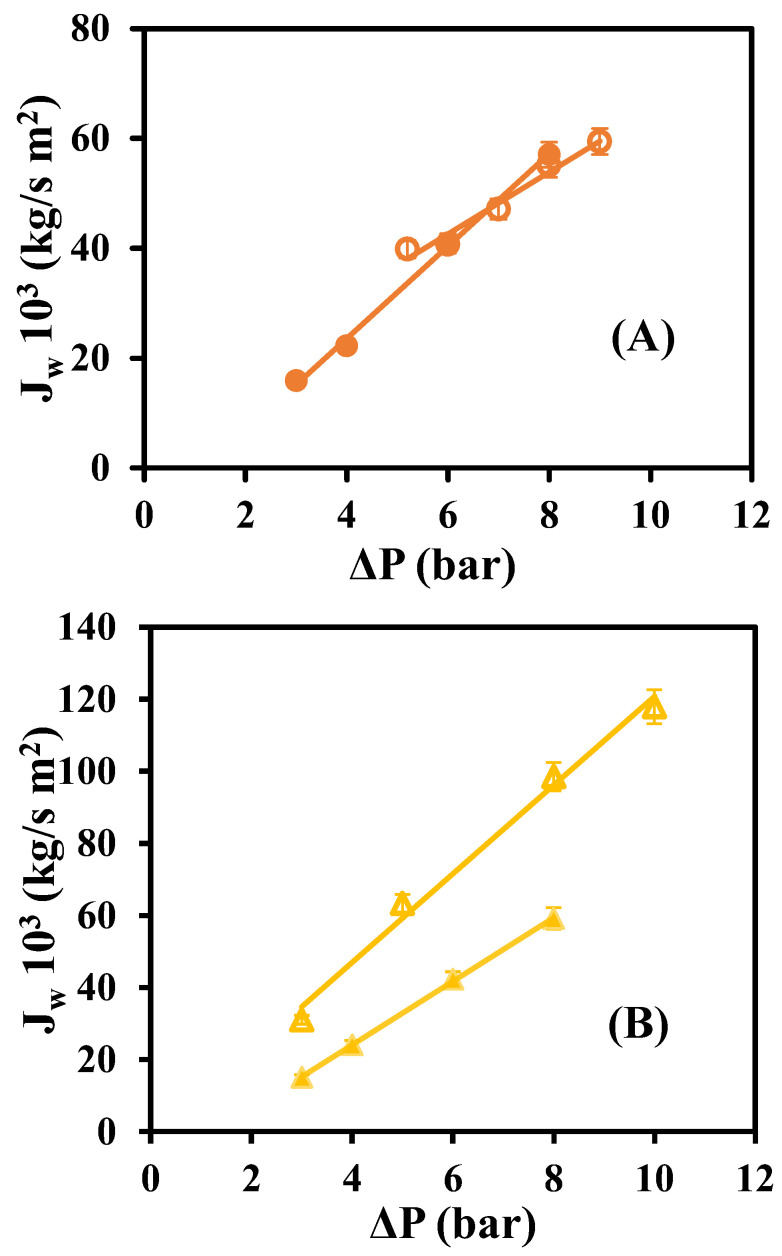
Water permeate flux versus operating pressures for native and rGO membranes: (**A**) GR60PP (
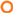
) native, (●) rGO-modified, (**B**) GR80PP (
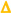
)) native (▲) RGO-modified.

**Figure 4 materials-16-01369-f004:**
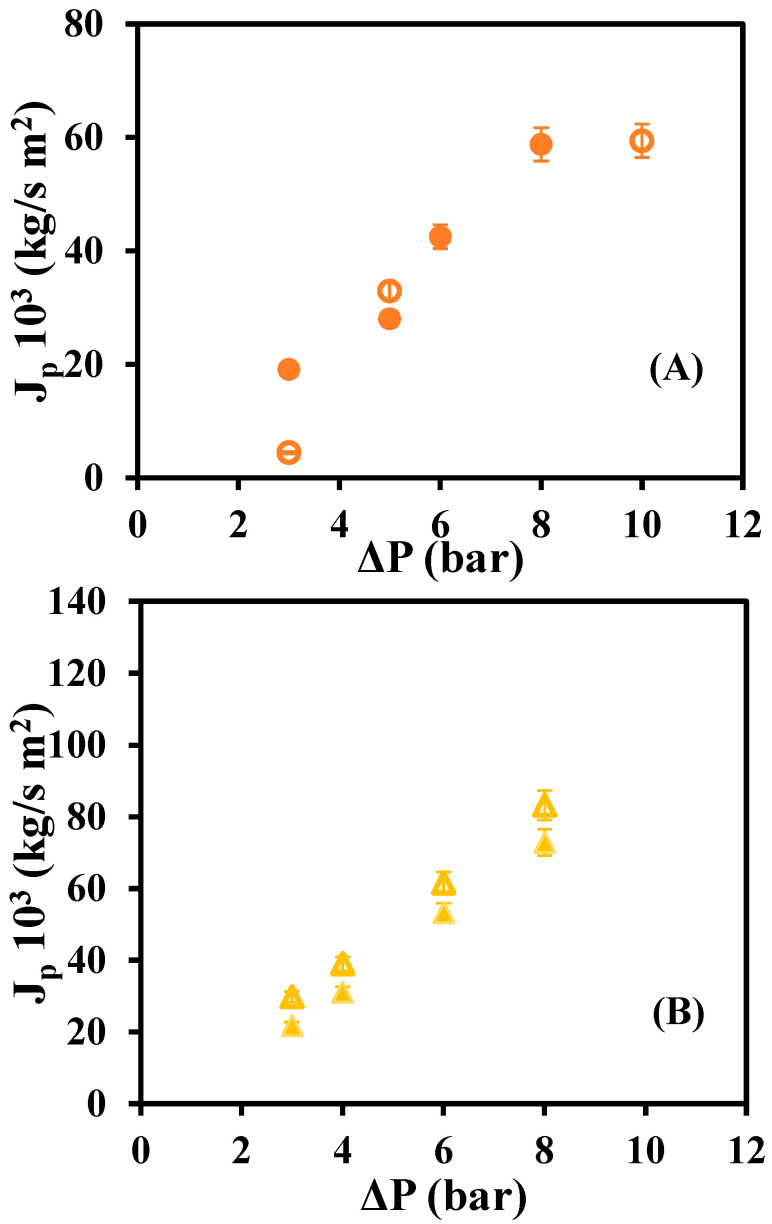
Dye permeate flux versus operating pressures for native and rGO membranes: (**A**) GR60PP (
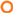
) native, (●) rGO-modified, (**B**) GR80PP (
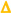
) native (▲) rGO-modified.

**Figure 5 materials-16-01369-f005:**
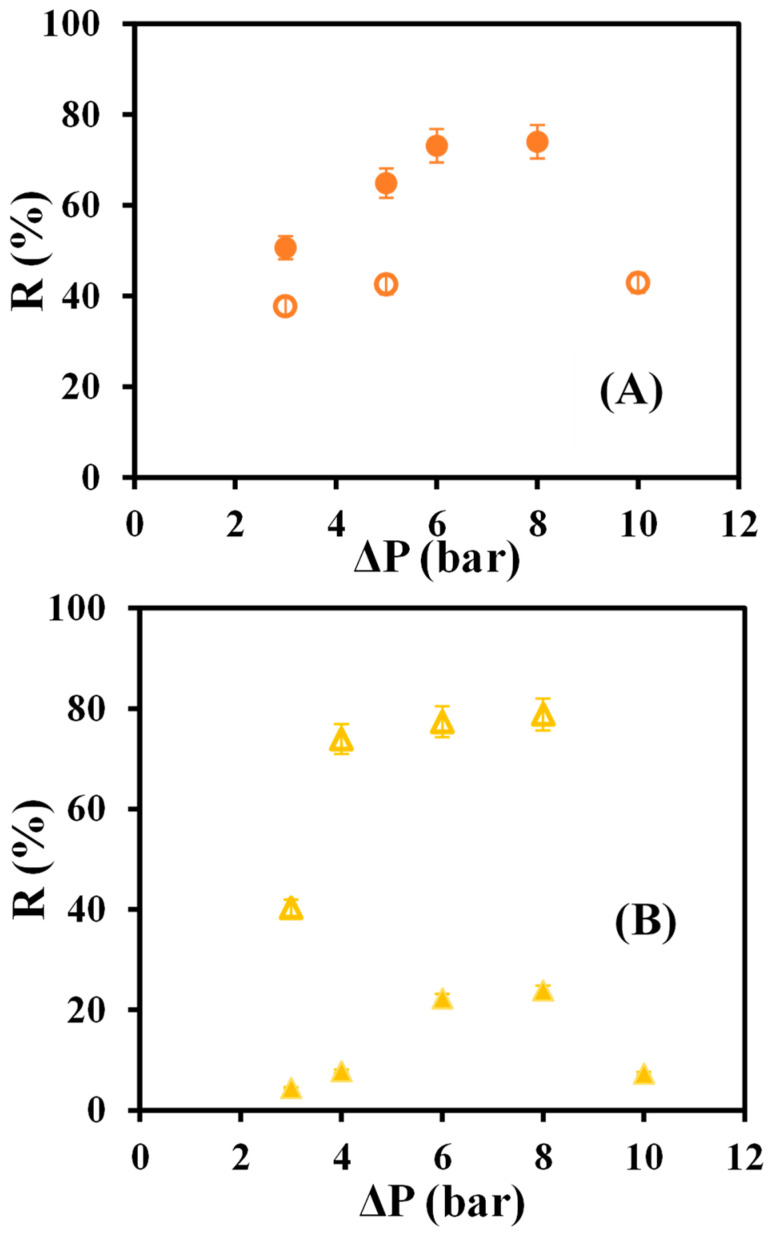
Rejection coefficient versus operating pressures for native and rGO membranes: (**A**) GR60PP (
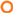
) native, (●) rGO-modified, (**B**) GR80PP (
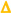
) native (▲) rGO-modified.

**Figure 6 materials-16-01369-f006:**
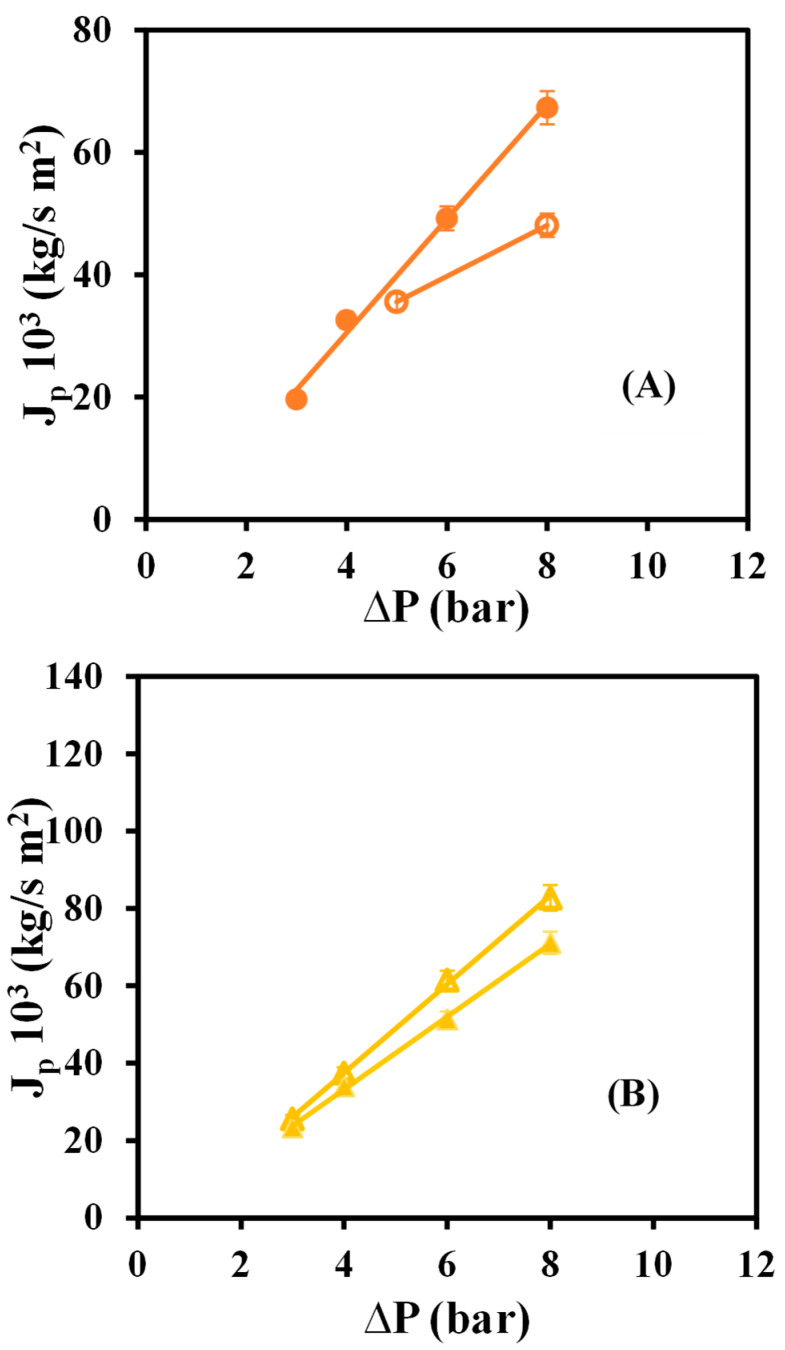
Water permeate flux versus operating pressures for native and RGO membranes: (**A**) GR60PP (
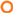
) native, (●)rGO-modified, (**B**) GR80PP (
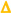
) native (▲) rGO-modified.

**Table 1 materials-16-01369-t001:** Main technical characteristics of ultrafiltration membranes.

Product Denomination	GR60PP	RC70PP	GR80PP
Manufacturer	Alfa Laval	Alfa Laval	Alfa Laval
Filtration type	Ultrafiltration	Ultrafiltration	Ultrafiltration
Molecular Weight Cut-Off (Da)	25,000	10,000	10,000
Composition	Polysulphone	Regenerated cellulose acetate	Polyethersulphone
Operating pressure range (bar)	1–10	1–10	1–10
Maximum pressure (bar)	10	10	10
pH range	2–10	2–10	2–10
Temperature range (°C)	5–70	5–70	5–70

**Table 2 materials-16-01369-t002:** Initial solvent permeability coefficients.

	Coefficient of Permeability to Solvent 10^8^ (s/m)	
Membranes	Native	RGO
GR60PP	6.41	8.41
GR80PP	12.31	8.85

**Table 3 materials-16-01369-t003:** Solvent permeability coefficients after dye-passing.

	Coefficient of Permeability to Solvent 10^8^ (s/m)	
Membranes	Native	RGO
GR60PP	4.16	9.30
GR80PP	11.47	9.42

## Data Availability

Not applicable.

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
