# Peer review of "Ultrafiltration Membranes Modified with Reduced Graphene Oxide: Effect on Methyl Green Removal from Aqueous Solution"

_materials, 2023, doi:10.3390/ma16041369_

Round 1

Reviewer 1 Report

In the present work, three commercially available membranes have been tested for the removal of the dye methyl green. For two of these membranes, the surfaces were modified with reduced graphene oxide. The conclusion is that one membrane (GR60PP) performs better if the surface is modified, and one performs better without surface modifications (GR80PP).  The other membrane is studied as provided by the manufacturer, without surface modification. In my opinion, the paper does not bring an important contribution to the field and needs major improvements before being reconsidered for publication. Below are a few aspects which attracted my attention.

1.      It is not clear to me why only 2 membranes (GR60PP and GR80PP) were modified if 3 membranes were considered in the study.

2.      It is not clear if surface modification worked. The SEM images in Figure 1 cannot be an argument. One can take images in different regions and see different roughness. A more comprehensive discussion of the results in Figure 2 would be necessary.  

3.      I do not see the connection of the present work with the statement between lines 97-101: “Since the synthesis of graphene oxide…solvents or binders”

4.      Table 1: The “pore size” is given in Da=Dalton. In fact, MWCO is in the unit of Dalton. In line 344 the “pore size” is mentioned again in kDa. Please check the terminology.

5.      Line 212: There is a comment in Spanish.

6.      Table 2: Permeability coefficients obtained with 3 decimals are not realistic. The data should be within the limit of experimental errors.

7.      Lines: 63, 256, 278: The word “experimentation” is inappropriate. It should be “experiment”.   

8.      On the abscissa axis in Figures 3,4,5,6 should be DeltaP.

9.      References 1 and 2 cannot be found. The link does not work. Also, such citations in a different language than English should be avoided. The communication language of the Journal is English.

10.   Reference 10. The title of the paper is missing.

11.   Reference 14 is in Spanish and is not accessible to most readers. 

Reviewer 2 Report

In this paper, selected membranes were modified with reduced GO, and the characteristics and performance were investigated. However, the experiments were not adequate to support the topic. After consulting sufficient literatures about the investigation of membrane technology for water treatment, such as membrane characterization, membrane operation, membrane fouling and membrane cleaning experiments, etc, the authors should redesign the experiments. In terms of the organization and phraseology, the readability and legibility of this manuscript should be improved and its novelty needs to be accurately emphasized. For clearness and concision, the manuscript should be rationally reorganized. Additionally, the manuscript should be polished by native speakers.

Specific comments:

Lines 42–44: “In recent years, ultrafiltration membrane processes have gained popularity because they are capable of removing most organic molecules, viruses and a wide variety of salts.”, Why ultrafiltration could remove salts? It seems confused.

Reviewer 3 Report

The authors present three types of ultrafiltration membranes with different characteristics (GR60PP, RC70PP and GR80PP), and , as reported in the abstract, with the aim to evaluate the removal of the dye methyl green. Althought some characterisation was done to better understand membrane behaviour, the aim of work is not well presented, the organization and the interpretation of the data is very confusing. 

Moreover typo of internal check has been reported in the submitted version. See page 5 line 34.

 For these reasons I do not suggest this paper to be published.

Reviewer 4 Report

The manuscript is well written, and the conclusions are supported by the experiments. But I have only one remark that should be explained and corrected before the publication. Through the manuscript, the names graphene oxide and reduced graphene are used interchangeably, i.e. authors are describing the influence of graphene oxide addition on the properties of membranes (lines 91-97) and then discussing the influence of reduced graphene oxide (line 100-101); in most of the manuscript the material is called reduced graphene oxide, but in line 293 it is called "graphene oxide". Because graphene oxide and reduced graphene oxide are different materials (reduced one has much lower concentration of oxygen functional groups) it should be clearly stated what material was used to modify the membrane. In would be good to add the information of C/O ratio of material used.

Round 2

Reviewer 1 Report

The authors have implemented the suggested changes and made the necessary corrections. I consider it acceptable for publication.

Reviewer 2 Report

The quality of this revised manuscript has been obviously improved. How did the results of Table 3 were obtained? The conditions and details of the fouling experiments should be provied in section 2 Material and Methods. In addition, the prospect and feasiblility for the application of the fabricated membranes should be pointed out.

Reviewer 3 Report

The quality of the paper was improved with respect to the previous version. There is a linear description and the experiments are more understandable.

It seems difficult to understand the last part of the paper, where the authors try to explain the different behavior observed for both membranes in relation to what was expected. 

In addition, the possible explanations are not clear, especially when the authors mention pores opening.

 I suggest to improve this last part of paper and then to submit again.
